# Co-Treatment of Erythroid Cells from β-Thalassemia Patients with CRISPR-Cas9-Based β^0^39-Globin Gene Editing and Induction of Fetal Hemoglobin

**DOI:** 10.3390/genes13101727

**Published:** 2022-09-26

**Authors:** Lucia Carmela Cosenza, Cristina Zuccato, Matteo Zurlo, Roberto Gambari, Alessia Finotti

**Affiliations:** 1Department of Life Sciences and Biotechnology, Section of Biochemistry and Molecular Biology, University of Ferrara, 44121 Ferrara, Italy; 2Center ‘Chiara Gemmo and Elio Zago’ for the Research on Thalassemia, University of Ferrara, 44121 Ferrara, Italy

**Keywords:** β-thalassemia, gene editing, CRISPR-Cas9, fetal hemoglobin induction, rapamycin

## Abstract

Gene editing (GE) is an efficient strategy for correcting genetic mutations in monogenic hereditary diseases, including β-thalassemia. We have elsewhere reported that CRISPR-Cas9-based gene editing can be employed for the efficient correction of the β^0^39-thalassemia mutation. On the other hand, robust evidence demonstrates that the increased production of fetal hemoglobin (HbF) can be beneficial for patients with β-thalassemia. The aim of our study was to verify whether the de novo production of adult hemoglobin (HbA) using CRISPR-Cas9 gene editing can be combined with HbF induction protocols. The gene editing of the β^0^39-globin mutation was obtained using a CRISPR-Cas9-based experimental strategy; the correction of the gene sequence and the transcription of the corrected gene were analyzed by allele-specific droplet digital PCR and RT-qPCR, respectively; the relative content of HbA and HbF was studied by high-performance liquid chromatography (HPLC) and Western blotting. For HbF induction, the repurposed drug rapamycin was used. The data obtained conclusively demonstrate that the maximal production of HbA and HbF is obtained in GE-corrected, rapamycin-induced erythroid progenitors isolated from β^0^39-thalassemia patients. In conclusion, GE and HbF induction might be used in combination in order to achieve the de novo production of HbA together with an increase in induced HbF.

## 1. Introduction

The β-thalassemias are hereditary pathologies caused, at the molecular level, by more than 300 mutations of the adult β-globin gene, leading to low or absent production of adult hemoglobin (HbA) in erythroid cells [1,2,3,4]. Together with sickle cell disease (SCD), the economic and clinical impacts of β-thalassemias are devastating in developing countries, where the frequency of these diseases is very high, mainly due to the lack of genetic counseling and prenatal diagnosis [1,2]. The therapeutic protocols for patients affected by β-thalassemia are currently based on blood transfusion, chelation therapy and, alternatively, bone marrow transplantation [3,4].

Within the clinical community, it is known that high blood content of fetal hemoglobin (HbF) is highly beneficial for patients with β-thalassemia [5,6,7], leading to milder forms of the disease. The earliest clinical observations indicating a key role of HbF in ameliorating the β-thalassemia phenotype came from patients with rare forms of β-thalassemia, particularly those with large deletions responsible for δβ^0^-thalassemia or the hereditary persistence of fetal hemoglobin (HPFH), characterized by the absence of β-globin and HbA in the presence of elevated levels of γ-globin chain production, resulting in high levels of HbF accumulation with a relatively benign clinical course [8]. More recently, clinical studies have shown that naturally elevated production of HbF improves the clinical course in a variety of β-thalassemia patients [9,10,11,12,13]. Accordingly, these observations have prompted research studies on HbF inducers that can therapeutically mimic, at least in part, what occurs in patients characterized by the natural persistence of high levels of HbF [14,15,16,17]. Some of these HbF inducers are presently considered in clinical trials (examples are NCT01245179, NCT00790127 and NCT03877809).

In addition to the approaches based on the pharmacological induction of fetal hemoglobin, an exciting strategy recently proposed for β-thalassemia is genome editing using a variety of protocols widely validated for hematopoietic cells [18]. In this specific field of investigation, the clustered regularly interspaced short palindromic repeats (CRISPR)-Cas9 nuclease system, is among the most efficient [19,20,21,22].

The possibility of using highly efficient gene-editing protocols opens new opportunities in the field of precision medicine for personalized therapy for β-thalassemia [23]. In this context, we have recently reported a protocol for the CRISPR-Cas9-based gene correction of the β^0^39-globin gene mutation (HGVS Name: HBB:c.118C > T), very frequent in the population of the Mediterranean area [24]. In addition to the precise correction of the β-thalassemia mutations, the CRISPR-Cas9-based gene editing approach has been extensively applied to the reactivation of HbF production in β-thalassemia erythroid cells, as demonstrated in the landmark work by Canver et al. [25]. In this case, the objective of the CRISPR-Cas9-based gene editing was (a) the silencing of transcriptional repressors of the γ-globin genes (such as BCL11A) [25,26,27,28,29,30] and (b) the disruption of their regulatory binding sites within the γ-globin genes, which in some cases mimics the natural HPFH mutations in the γ-globin gene [31,32,33,34,35,36,37]. For instance, Khosravi et al. demonstrated that the CRISPR-Cas9-based deletion of the BCL11A gene was associated with the reactivation of HbF production [26,27]. Of great interest is the fact that this strategy is currently under investigation in the NCT03655678 clinical trial, aimed at studying the safety and efficacy of CTX001 (hematopoietic cells gene-edited for elevated HbF production) on transfusion-dependent β-thalassemia (TDT) patients [29,38].

To maximize HbF production, these CRISPR-Cas9-based gene editing approaches can be combined, as recently proposed by Han et al. and by Samuelson et al. [39,40], who reported on a multiplex gene editing strategy based on the combination of two single gene editing approaches, one aimed at silencing the BCL11A repressor, the other aimed at disrupting the BCL11A binding sites present within the γ-globin gene promoter. Another example of multiplex gene editing approaches is that recently published by Psatha et al. [41], who studied the combination of CRISPR-Cas9-based cis and trans fetal globin reactivation mutations, demonstrating that this strategy leads to a significant increase in HbF production when comparison was performed with the single editing procedures. Accordingly, multiplex gene editing could be considered in clinical protocols finalized to the improvement of the clinical status of patients with a severe β-thalassemia phenotype. The results obtained in these studies concurrently demonstrated that these multiplex genomic editing protocols efficiently induced high levels of HbF production without increasing off-target effects [39] and without causing any defects in the proliferation rate or in the differentiation status of treated cells, either in vitro or in vivo [41].

The present study is aimed at determining whether HbF induction can be combined with the de novo production of HbA, obtained by the correction of a β^0^39-globin gene mutation using the CRISPR-Cas9 technology, as recently reported by Cosenza et al. [24].

To obtain co-production of increased levels of HbF and de novo synthesis of HbA, we first considered the possibility to perform CRISPR-Cas9-based multiplex genomic editing for BCL11A silencing (as reported by Khosravi et al., Frangoul et al. and Bjurström et al.) [27,29,30] and β^0^39 correction (as reported by Cosenza et al.) [24]. The advantage of this strategy is that both protocols use the same target cells (CD34^+^ erythroid progenitors) and no differences are expected in the clinical steps to be followed for collecting the CD34^+^ cells to be gene edited and for preparing the patients for the infusion of gene-edited cells (e.g., stem cells collected via mobilization and apheresis, myeloablative conditioning, infusion of corrected stem cells for the engraftment and immune reconstitution) [29]. On the other hand, a major drawback is expected, i.e., higher off-targeting effects and genotoxicity [40]. Supporting a caution in using multiplexed CRISPR-Cas9-based approaches, Samuelson et al. recently reported that multiplex CRISPR-Cas9 genome editing in hematopoietic stem cells for fetal hemoglobin reinduction generates chromosomal translocations [40]. For these reasons, we therefore decided to use for HbF induction a repositioned drug, rapamycin [42,43,44,45,46,47,48], among those already validated and used in clinical trials [49,50,51,52]. To the best of our knowledge, this strategy is novel, as no study is available on the combination of pharmacological induction of HbF with gene editing procedures aimed at the de novo production of HbA following the CRISPR-Cas9-based correction of genetic mutations.

Rapamycin, also known as sirolimus, is a potent inducer of HbF in in vitro systems [42,43,44,45,46,47,52], in in vivo animal models [46,47,53,54], and in few but highly informative patients affected by sickle-cell disease (SCD) [55,56]. In conclusion, all the available in vitro data concurrently indicate that rapamycin can be repurposed for the treatment of β-thalassemia for the following reasons: (a) rapamycin increases HbF in cultures from β-thalassemia patients with different basal HbF levels; (b) rapamycin increases the overall Hb content per cell; (c) rapamycin selectively induces γ-globin mRNA accumulation, with only minor effects on β-globin protein and β-globin mRNAs; (d) there is a strong correlation between the HbF increase induced by rapamycin and the increase in γ-globin mRNA content.

Accordingly, rapamycin is at present employed in two clinical trials recruiting β-thalassemia patients, NCT03877809 and NCT04247750 [57,58].

## 2. Materials and Methods

### 2.1. Isolation of Erythroid Precursor Cells (ErPCs) and ErPCs Cultures

ErPCs cultures were prepared from 25 mL of peripheral blood, following the Fibach protocol [59], as described by Zuccato et al. [58] and fully reported in the Appendix A. Immunological flow cytometry (FCM) characterization using antibodies for CD71 and CD235a demonstrated that the yield (% of ErPCs) was always higher than 85%, in agreement with previously reported data [58]. Representative FCM data and morphological analysis are shown in Appendix A. Elsewhere, published data demonstrate that the large majority of ErPCs undergo erythroid differentiation, as demonstrated with flow cytometry analysis using antibodies against transferrin receptor and glycophorin [60]. We carefully considered the fact that FBS might heavily affect ex vivo erythroid differentiation and hemoglobin production, thereby creating variability. For this reason, we screened all the FBS batches, selecting only those lacking effects on the differentiation of ErPCs and on HBF production after exposure to HbF inducers. Moreover, the same batch of FBS was used throughout all the experiments reported in the present study.

### 2.2. Treatment of Cells with β^0^39 CRISPR-Cas9 System and Rapamycin

On the third day of phase II, the ErPCs were considered ready to be treated with rapamycin, with β^0^39 CRISPR-Cas9 system or with the combined β^0^39 CRISPR-Cas9 system and rapamycin. Rapamycin (sirolimus, SIR, cat. R0395, Sigma Aldrich, St. Louis, MO, USA) was administered at the starting point of the ErPCs cultures at a concentration of 200 nM, and the stock solution was prepared by diluting the powder in EtOH 96% to reach a 50 μM concentration.

### 2.3. Cell Electroporation with CRISPR-Cas9 System for Correction of the β^0^39-Globin Gene Mutation

We followed the protocol described by Cosenza et al. [24], further detailed in Appendix A. Briefly, the genomic sgRNA target sequence was 5′-TGGTCTACCCTTGGACCTAG**AGG**-3′ (sgRNA target sequence underlined, PAM in bold); the gRNA complex begins by joining a tracrRNA (ATTO 550 labeled Alt-R^®^ CRISPR-Cas9 tracrRNA, IDT, USA), and the Alt-R^®^ CRISPR-Cas9 crRNA (IDT) oligonucleotide in thermoblock at 95 °C for 5 min.

### 2.4. Genomic DNA Extraction

The DNA was extracted from 200–300 µL of whole blood as described by Cosenza et al. [24] and detailed in the Appendix A. The quality of the genomic DNA was verified by gel electrophoresis using 0.8% agarose gels and quantified by spectrophotometry using the SmartSpec™ Plus instrument (Biorad Smartspec Plus, Bio-Rad).

### 2.5. RNA Isolation, cDNA Reverse Transcription and RT-qPCR

The total cellular RNA was extracted using the TRI Reagent^®^ (Sigma-Aldrich). After washing once with cold 75% ethanol, the RNA was dried and dissolved in diethylpyrocarbonate-treated water (WMBR: Water Molecular Biology Reagent nuclease-free, Sigma-Aldrich). For analysis of gene expression, 0.5 μg of total RNA was reverse transcribed by using the TaqMan^®^ Reverse Transcription Reagents and Random Hexamer (Applied Biosystems, Life Technologies, Thermo-Fisher, Waltham, MA, USA). The relative content of α-, β-, and γ-globin mRNAs were quantified by multiplex qPCR using primers and FAM, HEX and Cy5/ZEN/IBFQ-labeled hydrolysis probes purchased as custom-designed PrimeTime qPCR Assays from IDT and listed in Table 1.

Data of RT-qPCR experiments were analyzed using CFX Manager™ software (Bio-Rad). The relative expression of globins’ mRNAs was calculated using the comparative cycle threshold method (ΔΔCt method) using as reference genes human GAPDH sequences [24,58,61].

### 2.6. Droplet Digital PCR (ddPCR) to Evaluate Genomic and Transcriptomic β^0^39 Globin Correction

The evaluation of the β-globin gene correction levels in the position of codon 39 was carried out with droplet digital PCR [24,62]. In these experiments, Taq-Man probes marked with FAM and HEX fluorophores were used, designed specifically for the identification of the sequence containing the β^0^39 mutation (HEX) in the β-globin gene and the corresponding corrected sequence (FAM) (Table 1). The protocols have been reported by Cosenza et al. [24] and detailed in Appendix A.

### 2.7. HPLC Analysis of Hemoglobins

Analysis of HbA, HbF and free α-globin chains was performed with HPLC as elsewhere reported [24,58,61,63]. Lysates have been loaded into a PolyCAT-A cation exchange column and then eluted in a sodium-chloride-BisTris-KCN aqueous mobile phase using HPLC Beckman Coulter instrument System Gold 126 Solvent Module-166 detector, which allows to obtain for the quantification of the hemoglobins present in the sample. Further details can be found in Appendix A.

### 2.8. Western Blotting Analysis

The accumulation of β-globin (16 kDa) and γ-globin (15 kDa) proteins was assessed with Western blotting as described by Cosenza et al. [24] and detailed in Appendix A.

### 2.9. Amplicon Sequencing and Whole Genome Sequencing

All the experiments for the construction of the amplicon libraries, the sequencing of the fragments, and all the bioinformatics analysis (including the estimation of CRISPR-Cas9 OFF-target sites and analysis of OFF-target insertion) were performed at Genartis—Innovative Genomic Technologies laboratories (Genartis Srl, Verona, Italy, https://genartis.it/, accessed on 1 June 2022), following the same protocols reported in the previous work by Cosenza et al. [24].

### 2.10. Statistical Analysis

All the data are presented as mean ± S.D. Statistical differences have been determined using ANOVA (analyses of variance between groups) followed by Dunnett’s post hoc tests. Statistical differences were considered significant when *p* < 0.05, highly significant when *p* < 0.01 [58].

## 3. Results

### 3.1. Experimental Strategy for CRISPR-Cas9 Correction of the Thalassemia β^0^39 Mutation and for Co-Treatment with Rapamycin

Appendix A show the experimental strategy for the CRISPR-Cas9-based correction of the β^0^39-globin gene mutation in erythroid precursor cells (ErPCs) isolated from β-thalassemia patients and for the combination of this CRISPR-Cas9 treatment with rapamycin-based HbF induction. Rapamycin was used at 200 nM final concentration. ErPCs were first cultured for 7 days without erythropoietin (EPO) (Phase I). Then, the cells were transferred to a medium containing EPO (Phase II), for stimulating the erythroid differentiation and the production of hemoglobin. After three days of Phase II culture, the cells were electroporated in the presence of a reaction mix containing all the elements of the CRISPR-Cas9 system and/or treated with 200 nM rapamycin. After electroporation and genomic editing and/or rapamycin treatment, the cells were maintained in Phase II medium and, after 5 days, analyzed to evaluate the biological effects of the treatments. The immunophenotype of ErPCs and further details concerning morphology and key features of in vitro ErPCs differentiation are reported in Appendix A. In addition, key features of in vitro ErPCs differentiation have been reported elsewhere and discussed in the study published by Bianchi et al. [60]. In brief, the flow cytometry analysis of transferrin receptor (TR) and glycophorin A (GYPA) surface marker expression in ErPCs phase II cultures from β-thalassemia patients revealed GYPA as a late erythroid marker, with an increase from day 4 to day 8 along with Hb production, and TR as an early marker with unchanged expression from day 4, at >80% cells positive for either marker at both time points [60].

Concerning the analysis of gene editing, the used techniques allowed us to evaluate gene correction at the following levels: genomic (using sequencing and ddPCR protocols), transcriptomic (using RT-qPCR and RT-ddPCR approaches) and proteomic (using Western blotting and HPLC). Concerning the analysis of HbF induction, RT-qPCR and HPLC allowed us to compare the effects of the treatments on the accumulation of γ-globin mRNA and increased production of HbF, respectively.

The CRISPR-Cas9 model used for the correction of the β^0^39-globin gene mutation has been described by Cosenza et al. [24] and presented in Appendix A.

### 3.2. End-Point of the Gene Editing of the β^0^39-Globin Gene: Genomic Analyses and RT-ddPCR to Detect Corrected Normal β^0^39-Globin Gene and mRNA

In order to verify the presence of the normal β-globin gene after CRISPR-Cas9 correction of the β^0^39-thalassemia mutation, two complementary approaches were employed: (a) allele-specific PCR, performed using droplet-digital PCR and (b) amplicon sequencing. Figure 1A,B, shows a representative analysis of gene correction with the CRISPR-Cas9 system performed on ErPCs genomic DNA isolated from a homozygous β^0^39-thalassemia patient and cultured without treatments (−) or using the following experimental conditions: (a) CRISPR-Cas9 gene editing (GE); (b) 200 nM rapamycin (RAPA); (c) GE and rapamycin treatment (200 nM) (GE + RAPA). The correction data plotted are represented in the 1d dot plot, obtained from the ddPCR analysis software.

As is clearly evident, the presence of amplified edited β-globin gene sequences is absent in control untreated (−) and in rapamycin-treated (RAPA) ErPCs but present in both genome edited (GE) and edited + rapamycin-treated (GE + RAPA) cultures. The correction level obtained is reported as concentration (copies/µL of reaction) and in the form of fractional abundance %, calculated as an edited/edited + mutated concentration.

In Figure 1C,D the fractional abundance of corrected gene sequences is shown, obtained from four ErPCs populations. Results from amplicon sequencing (Appendix A) confirmed the editing of ErPCs.

Figure 1E,F shows a representative example of accumulation of β-globin mRNA using ErPCs from a β^0^39/β^0^39 homozygous β-thalassemia patient treated as described in Appendix A and analyzed by RT-ddPCR assay.

In Figure 1G,H, the data obtained from the same representative experiment are reported as concentration (copies/µL of reaction) and in the form of fractional abundance %, calculated as an edited/(edited + mutated) concentration. The fractional abundance data shown in Figure 1G,H demonstrate a high content of the edited β-globin mRNA in CRISPR-Cas9 treated ErPCs either in the absence or in the presence of the HbF inducer rapamycin.

Despite the fact that a direct translation from “fractional abundance” to “% of corrected cells” cannot be proposed, the data shown in Figure 1 clearly indicate that corrected β-globin gene (Figure 1A–D) and corrected β-globin mRNA (Figure 1E–H) are detectable only in gene-edited (GE) ErPCs populations. The high content of corrected β-globin mRNA is expected since it is well established that the β^0^39-globin mRNA (as most of mRNAs carrying stop-codon mutations) are highly unstable [24].

In order to evaluate the correction level of β^0^39-globin gene mutation obtained from ErPCs treated with our CRISPR-Cas9 system, we analyzed both mutated β^0^39 and edited β-globin mRNAs, using also an RT-ddPCR approach. γ-globin and α-globin transcripts were also analyzed.

### 3.3. De Novo Production of Edited β-Globin mRNA and Induction of γ-Globin Gene Transcription in the Same ErPCs Populations

Figure 2A,B reports a summary of the genomic and RT-ddPCR analyses to detect corrected normal β-globin mRNA, as well as the accumulation of α-globin, β-globin and γ-globin mRNA in the ErPCs analyzed obtained from different patients, comparing control untreated (−), with cells GE-corrected, cells treated with the HbF inducer rapamycin, and cells GE-treated and HbF induced. As expected, and in agreement with a previously published report from our laboratory [24] corrected β-globin gene sequences are present only in GE-treated cell populations, irrespectively to co-treatment with rapamycin (Figure 2A). In all the samples containing GE corrected β-globin gene sequences, the production of normal β-globin mRNA was readily detectable. The two ErPCs populations (GE and GE plus rapamycin) did not differ significantly with respect to the presence of the corrected β-globin gene (Figure 2A) and production of corrected β-globin mRNA (Figure 2B) (*p* = 0.2695 and *p* = 0.8910, respectively).

When the analysis was conducted for the content of α-, β-, and γ-globin mRNAs using RT-qPCR, the following results were obtained (Figure 2C–E). First of all, in agreement with Figure 2B, a significant increase in β-globin mRNA was detectable only in GE and GE plus rapamycin cultures (*p* = 0.0006 and *p* = 0.0089, respectively, with respect to rapamycin-only treated cultures) (Figure 2C,D). Importantly, when GE and GE plus rapamycin cultures were compared, no significant change in β-globin mRNA content was observed (*p* = 0.8286), demonstrating that rapamycin treatment has no major effects on β-globin mRNA content.

As a second and most relevant result, a significant increase in γ-globin mRNA was detectable only in rapamycin and GE plus rapamycin cultures (*p* = 0.0096 and *p* = 0.0152, respectively), with respect to GE-only treated cultures.

Moreover, when rapamycin and GE plus rapamycin cultures were compared, no significant change in the accumulation of γ-globin mRNA was observed (*p* = 0.5176), demonstrating that gene editing has no major effects on the rapamycin-mediated induction of the expression of γ-globin genes. Interestingly, no change in α-globin mRNA content was found in the ErPCs populations, indicating that the expression of α-globin genes in ErPCs treated with GE, rapamycin and GE plus rapamycin is similar to control untreated ErPCs (−). The data shown in Figure 2C–E were obtained using GAPDH sequences as internal reference. However, the same conclusion can be reached using RPL13A or β-actin internal controls (unpublished results).

### 3.4. Co-Production of HbA (De Novo) and HbF (Induced) in Gene-Edited ErPCs Treated with Rapamycin

We conclusively demonstrated the de novo production of HbA and the increased production of HbF in gene-edited, HbF-induced, ErPCs using HPLC. Representative HPLC analysis performed on CRISPR-Cas9 edited, rapamycin-induced ErPCs from three β^0^39/β^0^39-thalassemia patients are reported in Figure 3A–C.

The HPLC data, as clearly shown in the representative chromatograms, indicate a de novo production of adult hemoglobin (HbA) in all GE-samples analyzed, in agreement with the data indicating efficient gene editing (Figure 1 and Figure 2A,B). An increased production of HbF was in addition observed when the GE plus rapamycin cultures were compared with the GE-only samples. Figure 3D,E show the summary of the treatments performed in which the GE, rapamycin and GE plus rapamycin samples were compared with control untreated cells. The results obtained confirm that GE treatment does not interfere with HbF induction (Figure 3D). In addition, despite the fact that with the presence of the reactivation of γ-globin genes the β-globin gene expression might be analogically reduced, the data obtained demonstrate that the rapamycin induction of γ-globin genes does not interfere with the de novo HbA production using the GE approach (Figure 3E).

The conclusions of the HPLC studies were further confirmed by the Western blotting analyses shown in Figure 4, which aimed at evaluating the β-globin and γ-globin proteins produced under the different experimental conditions depicted in Figure 3. The amounts of CRISPR-Cas9-corrected β-globin protein (16 kDa) and γ-globin protein (15 kDa) were normalized with the quantity of housekeeping GAPDH (37 kDa) protein. The data obtained show that rapamycin, as expected, does not induce an increase in the β-globin protein in control cells; in addition, rapamycin treatment does not affect the accumulation of β-globin in CRISPR-Cas9 corrected ErPCs (Figure 4A).

On the other hand, the gene editing treatment has no effect on γ-globin accumulation, and, importantly, gene editing has no effect on rapamycin-mediated increase in γ-globin (Figure 4B). These data are consistent with the conclusion that the co-induction of β-globin and γ-globin proteins occurs in CRISPR-Cas9-edited, rapamycin-treated ErPCs.

### 3.5. Amplicon-Based and WGS Sequencing Results

Bar-coded amplicons were sequenced on a NovaSeq 6000 platform 150 phycoerythrin (PE) mode. The obtained number of fragments was ∼400,000 for the 18 (9 in duplicate) samples sequenced. The calculation of the frequency of the edited base and the indels at the sites of interest showed an editing rate between 7.1% and 8.8% for the edited base (chr11:5,226,774), with a deletion rate between 9.4% and 7.4%. A very low occurrence of insertions (lower than 0.1%) was detectable (Figure 5, Appendix A). All samples showed an editing rate above the control samples, generated as expected background values, indicating that the gene editing was efficient in all samples analyzed. The data show that, as expected and in agreement with the results published by Cosenza et al. [24] and with the data presented in Figure 5, Appendix A of the present paper, a consistent proportion of corrected sequences is present in all of the edited samples.

Interestingly, and fully in agreement with Figure 3A–C and Figure 4A, no corrected sequences are present in rapamycin-treated cells, and no further increase in corrected sequences is present in samples isolated from GE-ErPCs treated with rapamycin. Concerning indel effects, no insertions were found. On the contrary, deletions were detected in a proportion similar to the insertion of corrected sequences. Similar results were obtained in a WGS study (Appendix A, representative data), in which we calculated the frequency of the edited base and of the indels on the sites of interest. In particular, on the target site, no insertions were found, whereas there was a low number of deletions, in agreement with data obtained with the previously described amplicon sequencing approach.

## 4. Discussion

Gene editing with CRISPR-Cas9 technology is one of the most promising strategies to be exploited for the precise correction of hereditary mutations in a variety of monogenetic diseases. For instance, CRISPR-Cas9 has been employed in cystic fibrosis [64,65], sickle-cell disease [66,67], Huntington’s chorea [68], Duchenne muscular dystrophy [69,70], hemophilia [71,72], and chronic granulomatous disease [73].

Concerning thalassemia, CRISPR-Cas9 gene editing can be proposed for the efficient correction of the β^0^39-globin gene mutation (one of the most frequent in the Mediterranean area) recently reported by Cosenza et al. [24]. This approach was demonstrated to be able to force gene-edited cells to de novo produce HbA, with possible clinical advantages in case the protocol is used in clinical trials focusing on homozygous β^0^39-thalassemia patients.

On the other hand, robust evidence demonstrates that fetal hemoglobin (HbF) can be highly beneficial to β-thalassemia patients, leading to a milder phenotype and lower requirement of blood transfusions [27,31]. In this respect, several clinical trials with β-thalassemia and/or sickle-cell anemia patients are ongoing using HbF inducers, such as NCT01245179 (based on the HDAC inhibitor Panobinostat) [74], NCT00790127 (based on 2,2-dimethylbutyrate, HQK-1001) [50] and NCT03877809 (based on the mTOR inhibitor rapamycin) [58].

The aim of our study was to verify whether the de novo production of HbA using CRISPR-Cas9-based gene editing can be combined with HbF induction protocols. This idea is not new, as it was validated by Zuccato et al. using a combination of gene therapy using lentiviral vectors and HbF induction [75,76]. This strategy was deemed useful in consideration of the fact that while an increase in β-globin gene expression in β-thalassemia cells can be achieved with gene therapy, the de novo production of clinically relevant levels of adult Hb may be difficult to obtain. On the other hand, the fact that the increased production of HbF is beneficial in β-thalassemia, the combination of gene therapy and HbF induction appears to be a pertinent strategy for achieving clinically relevant results.

Combined treatment using gene editing and HbF induction approaches together have not been described so far. Our results conclusively demonstrate that gene editing and HbF induction might be used in combination in order to achieve the de novo production of HbA together with the increased production of induced HbF. In these combined treatments, mild conditions of gene editing can be used, thereby limiting off-targeting and genotoxic effects. These issues are important considering that GE in thalassemia and rapamycin treatment of β-thalassemia patients are both in clinical trials (see NCT03728322, NCT03655678, NCT05444894, NCT03877809 and NCT04247750). In this respect, it should be underlined that the approach here described based on combined treatments might be considered within the therapeutic field of personalized treatments in precision medicine. In this context, the CRISPR-Cas9-based correction of the β^0^39-thalassemia mutation can be applied to any patient with at least one β-globin allele carrying the β^0^39 mutation, such as β^0^39/β^0^39 homozygotes and compound heterozygotes for the β^0^39-globin gene. On the other hand, HbF induction can also be considered a personalized approach, as several gene polymorphisms (such as the XmnI) have been reported to be associated with high HbF induction levels [77,78,79].

In conclusion, the protocol here described is expected to be of interest to all clinicians working on hematological diseases, such as β-thalassemia, in particular for those working on β^0^-thalassemias. It should be underlined that also researchers working with sickle-cell disease (SCD) patients might be interested since, apart from the gene editing of the SCD locus, HbF is expected to be useful for SCD [49].

A limitation of this study is that no attempt has been made to fully characterize the biochemical/molecular targets of rapamycin. This should be done in future experimental efforts, as it will help in understanding some therapeutically relevant findings of our study. For instance, the absence of reciprocal regulation of the expression of γ- and β-globin genes is still remarkable, although the lack of inhibitory effects on β-globin gene expression in rapamycin-treated erythroid cells has already been reported [43,45,58]. In this respect, while a reciprocal decrease in β-globin was expected in HbF-producing cells [35], from the HPLC (Figure 3) and Western blotting (Figure 4) analysis, we do not see any reduction of HbA in RAPA- and GE-treated cells when comparison was conducted with GE-only treated cells. Analysis of the transcription machinery might be proposed for better understanding this finding. In addition, post-transcriptional effects cannot be excluded, considering the well-known effects of mTOR inhibitors on protein synthesis [80].

A second major limitation of our study is that we have not addressed in depth the effects of the treatment of gene-edited ErPCs populations with rapamycin apart from the changes in hemoglobin pattern found in the experiments reported in Figure 3 and Figure 4. This is an important point for identifying the required end-points of a possible future clinical trial based on the present study. In this respect, one of the issues to be considered is the effects of rapamycin treatment on the excess of free α-globin chains. In fact, rapamycin exhibits a very interesting effect, i.e., the decrease in this excess in vitro and in vivo [58], with the consequent reduction of the unbalanced α-globin/ β-like globin chain ratios [1]. This is a clinically relevant end point, since low α-globin protein expression is beneficial in β-thalassemia patients [81,82]. Interestingly, Lachauve at al. have demonstrated that the effect of rapamycin on the excess of free α-globin chains is caused by the ULK-1-dependent activation of autophagy [54]. It will be of interest to determine whether autophagy and decrease in the excess of free α-globin chains is activated in gene-edited rapamycin-treated ErPCs. To this end, ULK-1 mRNA and the autophagy-related p62, LC3-I/II proteins should be quantified in gene-edited rapamycin-treated ErPCs. Preliminary data obtained by HPLC analyses support the hypothesis that rapamycin treatment further reduces the free α-globin peak in gene-edited ErPCs (Appendix A).

## Figures and Tables

**Figure 1 genes-13-01727-f001:**
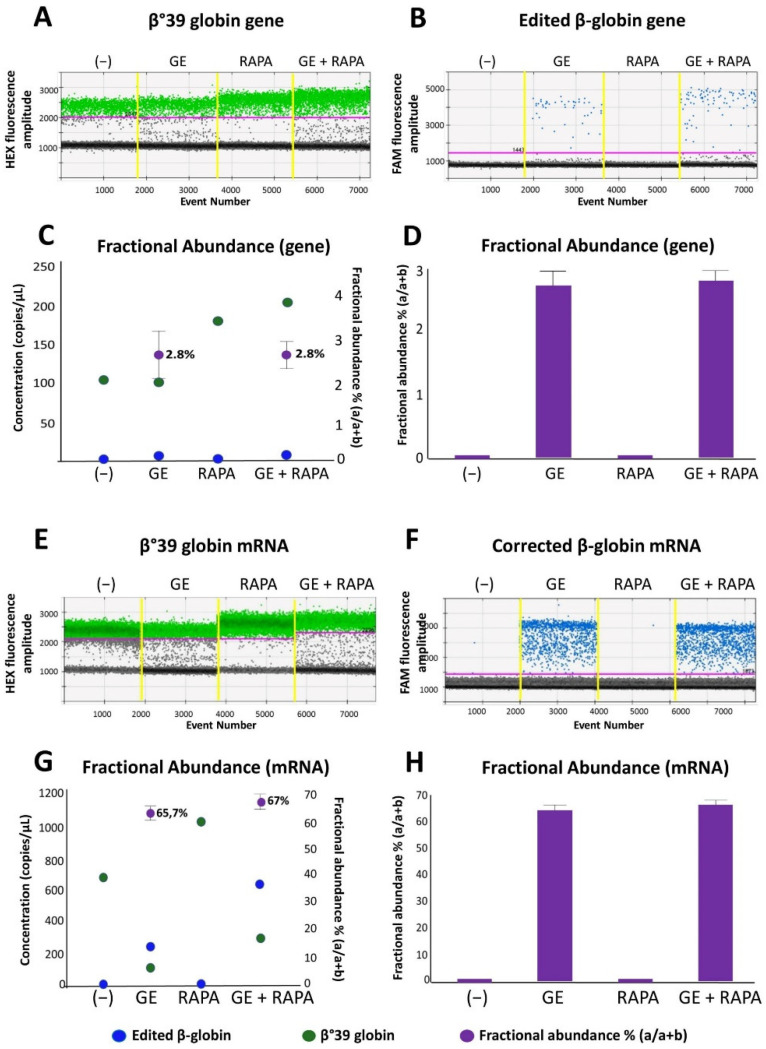
Evaluation of the effects of CRISPR-Cas9 system on β-globin gene and mRNA in ErPCs cultures. The panels show the data relating to a representative result obtained after gene correction treatments performed with the CRISPR-Cas9 system on a culture of β^0^39 ErPCs and analyzed with ddPCR assay. (**A**,**B**) and (**E**,**F**) refer to 1d dot plots obtained after analysis of mock treated (**A**,**E**—green dots) and edited (**B**,**F**—blue dots) β-globin gene and mRNA, respectively. (**C**,**G**) Correlation between the concentration of the samples (expressed in copies/µL) and the fractional abundance (purple dot) related to the representative example reported in panels (**A**,**B**) and (**E**,**F**). (**D**,**H**) Histograms extrapolated from the analysis of the fractional abundance of the representative sample used in the experiment. (−) control untreated cells; GE (cells treated with the CRISPR-Cas9 system), RAPA (rapamycin 200 nM) and GE + RAPA (cells treated with the CRISPR-Cas9 system and then cultured in the presence of rapamycin).

**Figure 2 genes-13-01727-f002:**
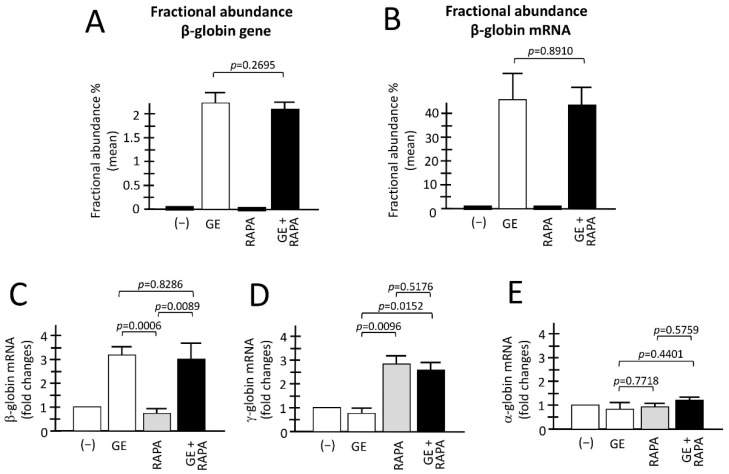
Evaluation of the combination between CRISPR-Cas9 gene editing and rapamycin-mediated HbF induction. (**A**,**B**) Fractional abundance obtained after treatment performed by CRISPR-Cas9 system on ErPCs isolated from β^0^39-thalassemia patients and analyzed by ddPCR assay. The histograms show the data related to β-globin gene (**A**) and β-globin mRNA (**B**). (**C**–**E**) The histograms show the relative content of the β-, γ- and α-globin mRNAs analyzed by multiplex RT-qPCR. (−): untreated cells; GE: cells treated with the CRISPR-Cas9 system; RAPA: rapamycin 200 nM treated cells; GE + RAPA: cells treated with the CRISPR-Cas9 system and cultured in the presence of rapamycin. All the data of RT-qPCR were normalized using GAPDH as housekeeping internal control gene, as described in Material and Methods. Results are expressed as fold changes with respect to control untreated cells (−). Results are from independent experiments using ErPCs cultures from three (**A**) and five (**B**–**E**) homozygous β^0^39-thalassemia patients. The level of statistical significance is reported as *p*-value (*p*).

**Figure 3 genes-13-01727-f003:**
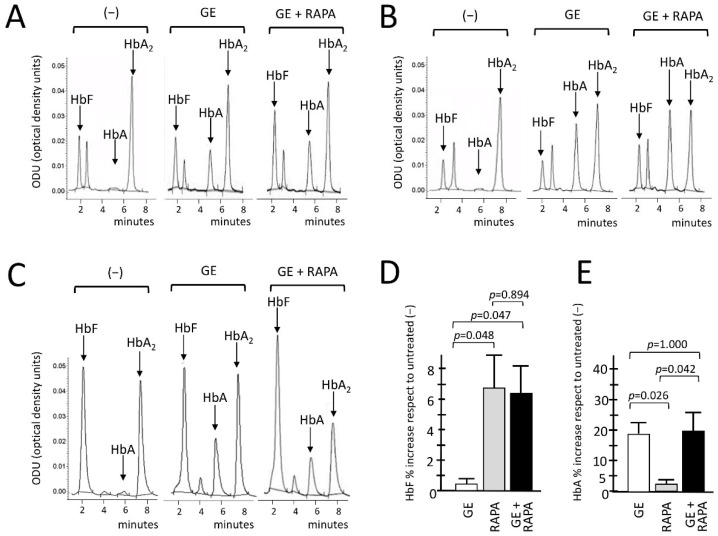
Effects of β^0^39 CRISPR-Cas9 treatment on HbA and HbF hemoglobins, evaluated with HPLC. (**A**–**C**) Chromatograms related to the HPLC analysis conducted of the protein lysates of the ErPCs cultures derived from three β^0^39-thalassemic patients. For each of them, the expression pattern of hemoglobins in ErPCs untreated (−), treated with β^0^39 CRISPR-Cas9 system alone (GE) and with the editing system in combination with rapamycin (GE + RAPA) were analyzed. The average of all patients analyzed for the relative increase in fetal hemoglobin HbF and adult hemoglobin HbA, expressed as a percentage, are shown in panels (**D**,**E**), respectively. The level of statistical significance is reported as *p*-value (*p*), significant when *p*< 0.05.

**Figure 4 genes-13-01727-f004:**
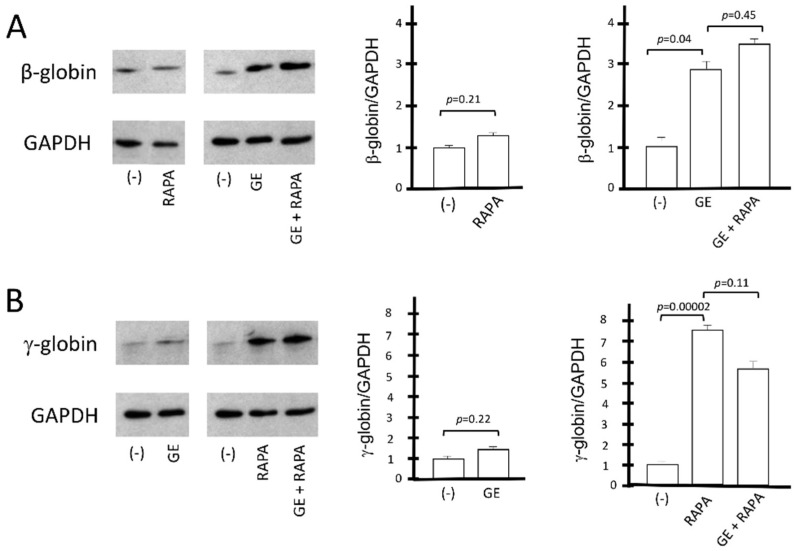
Western blotting analysis of β-globin protein and γ-globin protein. The relative content of β-globin (16 kDa) and γ-globin (15 kDa) proteins was determined by Western blotting analysis, after comparison with the housekeeping GAPDH (37 kDa) protein. (**A**) The effect on β-globin production of the β^0^39 CRISPR-Cas9 system is shown as representative data and graphically reported in the form of gel bands (panel (**A**), left). These results are reported in the right part of panel (**A**) as values of the densitometric analysis with respect to the reference protein GAPDH. (**B**) Representative data graphically show the impact of the β^0^39 CRISPR-Cas9 system on γ-globin expression. For the same samples, the data obtained from densitometric analysis of the Western blotting bands normalized on the housekeeping protein GAPDH (panel (**B**), left) are reported, and statistical significance is indicated as *p*-value (*p*). (−): untreated cells; GE (cells treated with the CRISPR-Cas9 system), RAPA (rapamycin 200 nM) and GE + RAPA (cells treated with the CRISPR-Cas9 system and then cultured in the presence of rapamycin). The original uncut versions of the gels are shown in Appendix A.

**Figure 5 genes-13-01727-f005:**
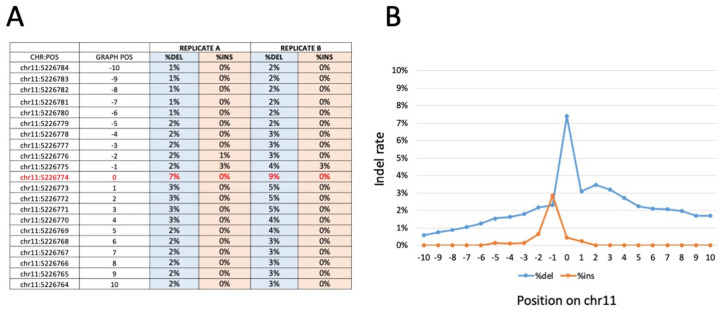
Representative indel frequency and positioning around the β^0^39 CRISPR-Cas9 editing target site, obtained with amplicon sequencing of β^0^39 ErPCs treated with the CRISPR-Cas9 system. Panel (**A**) shows the indel frequency values for the amplicon replicates analyzed. In particular, the ten positions upstream and downstream from position chr11: 5226.774 (β^0^39 target site indicated in red) were analyzed. The data obtained are reported in panel (**B**) in which the percentage of deletion (blue line) and insertion (orange line) is graphically reported as a function of each nucleotide position investigated. Similar results were obtained with WGS sequencing (Appendix A).

**Table 1 genes-13-01727-t001:** List of oligonucleotides (primers and probes) used to evaluate the correction degree obtained on the β-globin gene and study the accumulation of α-, β- and γ-globin mRNAs.

Oligonucleotides (Primers and Probes)	Sequence	Application
α-globin Forward Primer	5′-GGTCTTGGTGGTGGGGAAG-3′	RT-qPCR
α-globin Reverse Primer	5′-CGACAAGACCAACGTCAAGG-3′	RT-qPCR
α-globin Probe	5′-/5HEX/ACATCCTCT/ZEN/CCAGGGCCTCCG/3IABkFQ/-3′	RT-qPCR
β-globin Forward Primer	5′-GGTGAATTCTTTGCCAAAGTGAT-3′	RT-qPCR
β-globin Reverse Primer	5′-GGGCACCTTTGCCACAC-3′	RT-qPCR
β-globin Probe	5′-/5Cy5/ACGTTGCCCAGGAGCCTGAAG/3IAbRQSp/-3′	RT-qPCR
γ-globin Forward Primer	5′-TTCTTTGCCGAAATGGATTGC-3′	RT-qPCR
γ-globin Reverse Primer	5′-TGACAAGCTGCATGTGGATC-3′	RT-qPCR
γ-globin Probe	5′-/56-FAM/TCACCAGCA/ZEN/CATTTCCCAGGAGC/3IABkFQ/-3′	RT-qPCR
GAPDH Forward Primer	5′-TGTAGTTGAGGTCAATGAAGGG-3′	RT-qPCR
GAPDH Reverse Primer	5′-ACATCGCTCAGACACCATG-3′	RT-qPCR
GAPDH Probe	5′-/56-FAM/AAGGTCGGTCGGA/ZEN/GTCAACGGATTTGGTC/3IABkFQ/-3′	RT-qPCR
β-actin Forward Primer	5′-ACAGAGCCTCGCCTTTG-3′	RT-qPCR
β-actin Reverse Primer	5′-ACGATGGAGGGGAAGACG-3′	RT-qPCR
β-actin Probe	5′-/5Cy5/CCTTGCACATGCCGGAGCC/3IAbRQSp/-3′	RT-qPCR
β-glob Forward Primer	5′-CACTGACTCTCTCTGCCTATTG-3′	ddPCR β-globin gene
β-glob Reverse Primer	5′-ACC TTA GGG TTG CCC ATA AC-3′	ddPCR β-globin gene
β-glob β^0^39 Probe (HEX)	5′-/5HEX/TCTACCCTT/ZEN/GGACCTAGAGGTTCT/3IABkFQ/-3′	ddPCR β-globin gene
β-glob edit Probe (FAM)	5′-/56-FAM/TCTACCCTT/ZEN/GGACCCAGAGATTCT/3IABkFQ/-3′	ddPCR β-globin gene
β-glob Forward Primer	5′-TGGATGAAGTTGGTGGTGAG-3′	ddPCR β-globin mRNA
β-glob Reverse Primer	5′-CCTTAGGGTTGCCCATAACA-3′	ddPCR β-globin mRNA
β-glob β^0^39 Probe (HEX)	5′-/5HEX/TCTACCCTT/ZEN/GGACCTAGAGGTTCTT/3IABkFQ/-3′	ddPCR β-globin mRNA
β-glob edit Probe (FAM)	5′-/56-FAM/TCTACCCTT/ZEN/GGACCCAGAGGTTCTT/3IABkFQ/-3′	ddPCR β-globin mRNA

## Data Availability

Most of the data are included in the text and in the Appendix A. Additional information will be made freely available upon request to the corresponding authors.

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
