# Peer review of "Co-Treatment of Erythroid Cells from β-Thalassemia Patients with CRISPR-Cas9-Based β039-Globin Gene Editing and Induction of Fetal Hemoglobin"

_genes, 2022, doi:10.3390/genes13101727_

Round 1

Reviewer 1 Report

In this article LC Consenza et al. describe an attempt in combining different approaches to ameliorate the clinical severity of beta-thalassemia major. In a previous study the same group presented an efficient gene editing method to correct the β0CD39 genotype. Here, the authors suggest combining this approach with a simultaneous treatment of the patient cells with rapamycin, a known HbF inducer. Combining pharmacological induction of HbF with genomic modification of thalassemic and sickle hematopoietic cells has been suggested for a long time and data supporting the feasibility of this combination are of general interest.

Specific comments to the authors.

1)     It is known that by reactivating gamma globin, the beta globin gene expression is analogically reduced. Can the authors argument on why they chose to combine an HbF inducer such as rapamycin with a beta-globin gene correction approach rather than an HbF base or genome editing reactivation approach?

2)     The constitution of ErPCs is unclear from the materials and methods section. Does this cell population contain erythroid precursors, progenitors as well as HSPCs and myeloid cells which are abundant in the periphery of the thalassemic patients? Further characterization of the cell source (by FACS or morphological analysis) would be enlightening.

3)     Nowadays serum free conditions for the ex vivo erythroid differentiation is preferable since FBS might heavily affect hemoglobin production and create variability between different serum batches. FBS is present in all culture media presented here. Have the authors noticed such an effect? Was the same batch of serum used throughout these experiments?

4)     The figures of the paper can be consolidated. The experimental design is quite standard, therefore I find figure 1 redundant. Figures 2 and 3 can be combined in one. Figures 4,5 and 6  all show the effects on beta and gamma globin production. Combining them as well would ease the reader to compare the results between the different assays.

5)     Axes titles are missing from Figures 1 and 2 (I presume amplitude and event number).

6)     Since we do not expect rapamycin to induce a genomic correction, the comparisons of Figures 2 and 3 between the – and +RAPA samples are not informative and complicate the figure. I would suggest to simplify the figure by removing these groups.

7)     How does the “fractional abundance” in figure 2 translate to % of corrected cells?

8)     In Figure 4 C,D,E the axes titles are not descriptive and is not clear from the legend as to what this relative expression is being calculated.

9)     In Figure 5 A,B,C a control, untreated sample should be included.

10)  In general, the aim of this work is to show that the combination of gene correction with HbF induction is beneficial and could theoretically introduce an additive therapeutic effect. This hypothesis is not clearly supported by the results with maybe the exception of Figure 5D and E. Have the authors noticed any difference in the treated cells’ phenotype, proliferation and maturation? Results supporting their hypothesis, beyond the globin levels are required.

Author Response

Dear Reviewer #1,

Thank you very much for your work, comments and suggestions. We went through all the points raised, and found all your comments/suggestions very useful for the improvement of the scientific quality of our paper. We therefore emended the manuscript accordingly.

This is the list of the changes made (all of them are red-marked in the revised version, in order to facilitate your work) following all the points raised.

General comments

In this article LC Consenza et al. describe an attempt in combining different approaches toameliorate the clinical severity of beta-thalassemia major. In a previous study the same grouppresented an efficient gene editing method to correct the β0CD39 genotype. Here, the authors suggest combining this approach with a simultaneous treatment of the patient cells with rapamycin, a known HbF inducer. Combining pharmacological induction of HbF with genomic modification of thalassemic and sickle hematopoietic cells has been suggested for a long time and data supporting the feasibility of this combination are of general interest.

Answer. We thank the referee for her(his) comments and for the interest on our study.

Specific comments to the authors and point-by-point-reply.

Point 1. It is known that by reactivating gamma globin, the beta globin gene expression is analogically reduced. Can the authors argument on why they chose to combine an HbF inducer such as rapamycin with a beta-globin gene correction approach rather than an HbF base or genome editing reactivation approach?

Answer. We thank the reviewer for her/his comments and suggestions. Following the reviewer’s suggestion, we argumented in the revised version on why we chose to combine an HbF inducer (rapamycin) with a beta-globin gene correction approach rather than an HbF base or genome editing reactivation approach. The most important consideration to avoid multiplexed CRISPR-Cas9 approaches is that they are expected to cause genotoxicity [see the reference #40]. The most important point for selecting rapamycin as HbF inducer is that this repurposed drug is already in clinical trials (NCT03877809, A Personalized Medicine Approach for β-thalassemia Transfusion Dependent Patients: Testing sirolimus in a First Pilot Clinical Trial and NCT04247750, Treatment of β-thalassemia Patients with Rapamycin: From Pre-clinical Research to a Clinical Trial) [see reference #58]. In order to clarify this point the following sentence has been included: “In order to obtain co-production of increased levels of HbF and de novo synthesized HbA,…For these reasons, we therefore decided to use for HbF induction a repositioned drug, rapamycin [41−47], among those already validated and used in clinical trials [48−52]” (page 2 lines  86−98 and page 3, lines 99−101). To support the above-mentioned sentences, the following references have been added: 27, 29, 30, 40−52. Finally, we briefly commented on the possibility that “by reactivating gamma globin, the beta globin gene expression is analogically reduced”. This was done by including the following statement. “In addition, despite the fact that with the presence of the reactivation of γ-globin genes the β-globin gene expression night be analogically reduced, the data obtained demonstrate that rapamycin induction of γ-globin genes does not interfere with de novo HbA production using the GE approach (Figure 3E and data not shown). “ (page 11, lines 444−448).

Point 2. The constitution of ErPCs is unclear from the materials and methods section. Does this cell population contain erythroid precursors, progenitors as well as HSPCs and myeloid cells which are abundant in the periphery of the thalassemic patients? Further characterization of the cell source (by FACS or morphological analysis) would be enlightening.

Answer. We thank for raising this point. We have included in the Supplementary material section two Figures, one on the ErPC immunophenotype (Figure S3), the other on ErPC morphology (Figure S4). In addition, on the basis of elsewhere published observations [new refs. #58 and #60], we know that the majority of the ErPC population (>80-85%] undergo erythroid differentiation, on the basis of FACS analysis using antibodies against transferrin receptor and glycophorin A. The following sentence has been added to clarify this point: “The yield in terms of percentage of ErPCs was always higher than 85%, as suggested by immunological FACS… the large majority of ErPCs undergo erythroid differentiation, as demonatrated by FACS analysis using antibodies against transferrin receptor and glycophorin [60]” (page 3, lines 147−150 and page 4, lines 151−153). Moreover, we have commented this issue also in the Results section, including the sentence “The immunophenotype of ErPCs and further details concerning morphology and key features of in vitro ErPCs differentiation have been reported… GYPA+ cells is very high both at day 4 and day 8 of ErPC culture (>80%) [60]”, at page 7, lines 285−299.

Point 3. Nowadays serum free conditions for the ex vivo erythroid differentiation is preferable since FBS might heavily affect hemoglobin production and create variability between different serum batches. FBS is present in all culture media presented here. Have the authors noticed such aneffect? Was the same batch of serum used throughout these experiments?

Answer. We have not used serum-free conditions. However, the FBS used has been always pre-tested to verify lack of effects on erythroid induction of ErPCs. In addition, the same batch of pre-tested serum has been used in all the experiments performed. In order to clarify this point, the following sentence has been added: “We carefully considered the fact that FBS might heavily affect ex vivo erythroid differentiation and hemoglobin production, thereby creating variability… Moreover the same batch of FBS was always used throughout all the experiments reported in the present study.” (page 4, lines 153−158).

Point 4. The figures of the paper can be consolidated. The experimental design is quite standard,therefore I find figure 1 redundant. Figures 2 and 3 can be combined in one. Figures 4,5 and 6 all show the effects on beta and gamma globin production. Combining them as well would ease the reader to compare the results between the different assays. Answer. In order to follow these suggestions, these actions have been undertaken: (a) We moved Figure 1 in Supplementary materials [now this Figure is Figure S1; (b) We have combined Figure 2 and Figure 3 to generate one single Figure (Figure 1 of the revised version of the paper). We would like to maintain Figures 4, 5 and 6 (some of which have been revisited and implemented) as separate figures, in order to facilitate the readers, avoiding the inclusion in the paper of excessively complex figures. These figures become Figures 2, 3 and 4 of the revised manuscript.

Point 5. Axes titles are missing from Figures 1 and 2 (I presume amplitude and event number).

Answer. Done. We have added the axes titles.

Point 6. Since we do not expect rapamycin to induce a genomic correction, the comparisons of Figures 2 and 3 between the – and +RAPA samples are not informative and complicate the figure. I would suggest to simplify the figure by removing these groups.

Answer. In order to meet the reviewer’s suggestion, the control untreated (-) and the + RAPA sample groups are not present in most of the Figures. We would like to maintain these groups in the new Figures 1, 2 and 3, as we believe that the demontration that rapamycin treatment does not lead to increase of beta-globin mRNA is important, No gene correction, neither read-through activities are therefore present in RAPA-treated cells. (This was commented with the inclusion of the sentences “Importantly, when GE and GE plus rapamycin cultures are compared, no significant change in β-globin mRNA content was observed (p = 0.8286), demonstrating that rapamycin treatment has no major effects on β-globin mRNA content.”, at page 9, lines 390−392).

Point 7. How does the “fractional abundance” in figure 2 translate to % of corrected cells?

Anwer. We cannot state that fractional abundance correspond to a % of corrected cells. In order to clarify this point, the following sentence has been included: “Despite the fact that a direct translation from “fractional abundance” to “% of corrected cells” cannot be proposed, the data shown in Figure 1 clearly indicate… since it is well established that the β039-globin mRNA (as most of mRNAs carrying stop-codon mutations) are highly unstable [24].” (page 9, lines 361−366). 

Point 8. In Figure 4 C,D,E the axes titles are not descriptive and is not clear from the legend as to what this relative expression is being calculated.

Answer. Figure 4 is now renumbered as Figure 2 of the revised manuscript. For clarification, following the reviewer’s suggestion, we have modified the Y-axes of panels C, D and E, and included a description of the approach used for calculating the relative expression, in the legend of Figure 2 and in the Materials and Methods section, in the paragraph “2.5. RNA isolation, cDNA Reverse transcription and RT-qPCR” (page 5, lines 199−203).

Point 9. In Figure 5 A,B,C a control, untreated sample should be included.

Answer. Done. We have included in panels A, B and C the chromatograms relative to control untreated cells (−).

Point 10. In general, the aim of this work is to show that the combination of gene correction with HbF induction is beneficial and could theoretically introduce an additive therapeutic effect. This hypothesis is not clearly supported by the results with maybe the exception of Figure 5D and E. Have the authors noticed any difference in the treated cells’ phenotype, proliferation and maturation? Results supporting their hypothesis, beyond the globin levels are required.

Answer. This is a vary important point and a limit of our study. Among the possibility to study activation of rapamycin-assoaciated parameters beyong the globin levels, one of the most interesting is in our opinion the rapamycin-mediated activation of autophagy leading to a clinical relevant effects, i.e. decrease of the excess accumulation of free-alpha globin peak. To follow the reviewer’s suggestion the following sentence has been included: “The major limitation of our study is that we have not addressed in deep the effects…. support the hypothesis that rapamycin treatment further reduces the free α-globin peak in gene-edited ErPCs (Figure S8)”, page 14, lines 553−570).  Preliminary data on the effects of rapamycin on GE ErPCs were presented in Supplementary material (Figure S8), supporting the hypothesis of a beneficial effect of rapamycin (i.e. decrease of free alpha-globin chains) beyong changes in hemoglobin patters (i.e. increase of HbF).

In conclusion, we hope hat the changes included will be considered acceptable for answering to all the comments raised. Waiting for you comments we aoul like to thank again for your constructive and excellent work, which was judged by us to be very important to help us to implement the scientific quality of the paper.

Best regards and thanks again,

Dr. Alessia Finotti,

Department of Life Sciences and Biotechnology, Biochemistry and Molecular Biology Section, Ferrara University, Via Fossato di Mortara 74, 44121 Ferrara, Italy.

Tel: +39-0532-974510, Fax: +39-0532-974500, e-mail: [email protected]

Prof. Roberto Gambari,

Department of Life Sciences and Biotechnology, Biochemistry and Molecular Biology Section, Ferrara University, Via Fossato di Mortara 74, 44121 Ferrara, Italy.

Tel: +39-0532-974443, Fax: +39-0532-974500, e-mail: [email protected]

Reviewer 2 Report

In this article, Cozensa et al. utilize CRISPR-Cas9 gene editing technique for correcting the β039 thalassemia mutation in erythroid progenitors and demonstrates the induction of HbA and HbF. The method is validated in several ways and the analyses seems sound and the observations are potentially beneficial for future therapeutic relevance in hematological disorders like SCD and thalassemia. Overall, the findings are well presented, however below are some minor comments/points that the authors may wish to address.

Comments

1. One importance of this method is the ability to convey future studies to better identify factors affecting off target effects. I would challenge the authors to include this in the discussion of how this approach may be utilized to observe the effects outside of the gRNA sequence.

2. Line 24:  To remove “(4)” from the abstract

3. Line 52-57:  Authors needs to re-format sentence.

4. Please change:  De-novo, in vitro, in vivo (all needs to be italicized)

5. Line 101: Prefixes/factors needs to be corrected:  Dexamethasone 10−6M (Sigma Genosys), 10−5M β-mercaptoethanol

6. Non-uniformity in describing Rapamycin:  Change Sirolimus to RAPA (Figure 1)

7. Line 443-446:  In the discussion include few sentences and suitable references for Gene editing and HbF induction experiments performed by others, for example, Canver et.al BCL11A enhancer dissection by Cas9-mediated in situ saturating mutagenesis and Wu et al. Highly efficient therapeutic gene editing of human hematopoietic stem cells.

8. Line 208: Dunnett's was misspelled.

9. The authors must insert space between numbers and units such as rapamycin 200nM.

Author Response

Dear Reviewer #2,

Thank you very much for your work, comments and suggestions. We went through all the points raised, and found all your comments/suggestions very useful for the improvement of the scientific quality of our paper. We therefore emended the manuscript accordingly.

This is the list of the changes made (all of them are red-marked in the revised version in order to facilitate your work) following all the points raised.

General comments. In this article, Cosenza et al. utilize CRISPR-Cas9 gene editing technique for correcting the β39thalassemia mutation in erythroid progenitors and demonstrates the induction of HbA and HbF.The method is validated in several ways and the analyses seems sound and the observationsare potentially beneficial for future therapeutic relevance in hematological disorders like SCD and thalassemia. Overall, the findings are well presented, however below are some minor comments/points that the authors may wish to address.

Answer. We thank the reviewer for these encouraging comments.

Specific points.

Point 1. One importance of this method is the ability to convey future studies to better identify factors affecting off target effects. I would challenge the authors to include this in the discussion of how this approach may be utilized to observe the effects outside of the gRNA sequence.

Answer: In order to better explain the interest of ou study in the context of possible future clinical trials the following sentence has been included: “In these combined treatments, mild conditions of gene editing can be used, therefore limiting off-targeting and genotoxic effects. These issues are important considering that GE in thalassemia and rapamycin treatment of -thalassemia patients are both under in clinical trials (see NCT03728322, NCT03655678, NCT05444894, NCT03877809 and NCT04247750)” (page 14, lines 535−539).

Point 2. Line 24: To remove “(4)” from the abstract

Answer. Done.

Point 3. Line 52-57: Authors needs to re-format sentence.

Answer. Done. We have extensively changed and implemented this sentence,in the introduction section, in order to meet this and other reviewers’ comments (page 2; lines 81−85). 

Point 4. Please change: De-novo, in vitro, in vivo (all needs to be italicized)

Answer. Done. We have italicized these words throughout the text.

Point 5. Line 101: Prefixes/factors needs to be corrected: Dexamethasone 10−6M (Sigma Genosys),10−5M β-mercaptoethanol

Answer. Done. Sorry for the typo mistakes.

Point 6. Non-uniformity in describing Rapamycin: Change Sirolimus to RAPA (Figure 1)

Answer. We have changed “Sirolimus” to Rapamycin” in figure 1 (new Figure S1) and where it was cited in the main text.

Point 7. In the discussion, include few sentences and suitable references for Gene editing and HbF induction experiments performed by others, for example, Canver et.al BCL11A enhancer dissection by Cas9-mediated in situ saturating mutagenesis and Wu et al. Highly efficient therapeutic gene editing of human hematopoietic stem cells.

Answer. We thank you for this suggestion. We agree that citation of the work done by others is a required step in all publications, and we are sorry for the lack of several citations in the originally submitted manuscript. In order to give the most extensive description of previously published studies on gene editing for HbF induction, and the highest visibility, the following sentence has been included in the introduction of the manuscript: “The possibility to use highly efficient gene-editing protocols opens new avenues in the fields of personalized treatment and precision medicine of β-thalassemia [23]…. Interestingly, no increase of off-target effects have been reported [39].” (page 2  lines 55−80). The most relevant studies on  CRISPR-Cas9-based gene editing strategies for (a) the silencing of transcriptional repressors of the γ-globin genes (such as BCL11A) [25−30], (b) the disruption of their regulatory binding sites within the γ-globin genes [31−35], and (c) the induction of natural hereditary persistence of fetal hemoglobin mutations in the γ-globin gene [36,37], have been included. All the studies mentioned by the reviewer have been included (new references #23 and #28).

Point 8. Line 208: Dunnett's was misspelled.

Answer. We have corrected this typo mistake. Thanks.

Point 9. The authors must insert space between numbers and units such as rapamycin 200nM. Answer. Done.

In conclusion, we hope hat the changes included will be considered acceptable for answering to all the comments raised. Waiting for you comments we would like to thank again for your constructive and excellent work, which was judged by us to be very important to help us to implement the scientific quality of the paper.

Best regards,

Dr. Alessia Finotti,

Department of Life Sciences and Biotechnology, Biochemistry and Molecular Biology Section, Ferrara University, Via Fossato di Mortara 74, 44121 Ferrara, Italy. Tel: +39-0532-974510, Fax: +39-0532-974500, e-mail: [email protected]

Prof. Roberto Gambari,

Department of Life Sciences and Biotechnology, Biochemistry and Molecular Biology Section, Ferrara University, Via Fossato di Mortara 74, 44121 Ferrara, Italy. Tel: +39-0532-974443, Fax: +39-0532-974500, e-mail: [email protected]

Round 2

Reviewer 1 Report

The authors have satisfactory addressed all my comments/questions in my initial review. The revised manuscript is easier to follow and better highlights the rationale of the study and importance of the findings. I have only two minor comments. Regarding the multiplexed genome editing discussed in the introduction the authors have overlooked the first application, published in 2021 (Psatha N, Georgakopoulou A, Li C, et al. Blood. 2021;138(17):1540-1553),  describing the method in depth in both normal and thalassemic CD34+ cells, in vitro and in vivo, which is more relevant to the present paper. Also the phrase "and data not shown" is better to be removed from lines 446, 447 or the data to be added to the figure or supplement.

Author Response

Dear Reviewer #2,

Thank you very much for your further suggestions. We amended the manuscript accordingly.

This is the list of the changes made (all of them are red-marked in the revised version in order to facilitate your work) following all the points raised.

General comments. The authors have satisfactory addressed all my comments/questions in my initial review. The revised manuscript is easier to follow and better highlights the rationale of the study and importance of the findings. I have only two minor comments.

Answer. We thank the reviewer for her(his) comments and the points raised.

Point 1. Regarding the multiplexed genome editing discussed in the introduction the authors have overlooked the first application, published in 2021 (Psatha N, Georgakopoulou A, Li C, et al. Blood. 2021;138(17):1540-1553), describing the method in depth in both normal and thalassemic CD34+ cells, in vitro and in vivo, which is more relevant to the present paper.

Answer. We are grateful for this suggestion, which allows the “Introduction” section to be far more complete. We have included the suggested study by Psatha et al. (Reference #41 of the R2 revised version). We have also briefly commented on the results of the paper by including the sentence “Another example of multiplex gene editing approaches is that recently published by Psatha et al. [41], who reported……proliferation and differentiation of treated cells, either in vitro or in vivo [41]” (page 2, lines 77−86).

Point 2. Also the phrase "and data not shown" is better to be removed from lines 446, 447 or the data to be added to the figure or supplement.

Answer. We have removed the phrase, according to your suggestion.

In conclusion, we hope that the changes included will be considered acceptable for answering all the comments raised.

Waiting for your comments we would like to thank you again for your constructive and excellent work, very important t in helping us to implement the scientific quality of the paper.

Best regards,

Dr. Alessia Finotti, Department of Life Sciences and Biotechnology, Biochemistry and Molecular Biology Section, Ferrara University, Via Fossato di Mortara 74, 44121 Ferrara, Italy. Tel: +39-0532-974510, Fax: +39-0532-974500,                      e-mail: [email protected]                                                       

Prof. Roberto Gambari, Department of Life Sciences and Biotechnology, Biochemistry and Molecular Biology Section, Ferrara University, Via Fossato di Mortara 74, 44121 Ferrara, Italy. Tel: +39-0532-974443, Fax: +39-0532-974500, e-mail: [email protected]
